# Analysis and Design of Lateral Framing Systems for Multi-Story Steel Buildings

Husam Al Dughaishi [1,2,*], Jawad Al Lawati [1], Moad Alosta [1], Shaker Mahmood [3], Mohamed Faisal Al-Kazee [4], Nur Izzi Md Yusoff [5] and Abdalrhman Milad [1,*]

[1] Department of Civil and Environmental Engineering, College of Engineering, University of Nizwa, P.O. Box 33, Nizwa 616, Oman

[2] School of Architectural Engineering and Construction, University of Nebraska–Lincoln, 130R Whittier Bldg., Lincoln, NE 68583, USA

[3] Department of Civil Engineering, College of Engineering, University of Duhok (UoD), P.O. Box 78, Duhok 1006, Iraq

[4] Department of Architecture and Interior Design, College of Engineering and Architecture, University of Nizwa (UoN), P.O. Box 33, Nizwa 616, Oman

[5] Department of Civil Engineering, Universiti Kebangsaan Malaysia, Bangi 43600, Malaysia

[*] Correspondence: husam@unizwa.edu.om (H.A.D.); a.milad@unizwa.edu.om (A.M.)

**Abstract:** This study focused on identifying the most appropriate structural system for multi-story buildings and analyzing its response to lateral loads. The study analyzed and compared the different structural systems to determine the most suitable option. The study aims to utilize three lateral framing systems (moment, braced, and diagrid) in order to investigate which system needs the least amount of steel to meet the design requirements. Thus, in order to determine the estimated steel savings of this system as compared to the moment and braced frames, the four-story and eight-story buildings that are $96' \times 96'$ in the plane and utilize moment frames, braced frame, and diagrid framing structural systems are presented. Based on the American Society of Civil Engineers (ASCE) 7–10, load combinations are considered for the designs, and the RAM structural analysis is used for the modeling and analysis of the structural systems. The findings of this study's illustrations were the optimum for the analysis of wind of 176 kips and seismic loads of 122 kips, the building's lateral displacements, which were the lowest at 0.045 inches, the story drift, the story stiffness, and the story shear for each structural system. In addition, the diagrid system also had the least amount of shear for all the stories, suggesting that it is better able to manage the lateral forces. These results indicate that the diagrid system is a more efficient structural system and can be recommended for use in multi-story buildings.

**Keywords:** steel buildings; multi-story buildings; lateral framing systems; diagrid system; moment frame





## 1. Introduction

### 1.1. Research Background

During the late 19th and early 20th centuries, engineering development created thin shell and tensile structures of extraordinary refinement and elegance, using the diagrid system with various types, such as the metal, concrete, or wooden beams that are used in the construction of buildings [1]. However, their studies were based on the elastic and plastic behavior of simple rectangular uniform diagonal grids, and the studies tried to find formulas for their collapse loads [2,3]. Due to the rapidly growing population and the scarcity of urban land, the last few decades have required architectures that must consider ways of enabling vertical growth. Structural safety has always been a key preoccupation and responsibility in the design of civil engineering projects [4]. Steel framing systems typically refer to a building technique such as vertical steel columns and horizontal I-beams.

However, the framing system used in a building works by transferring all building loads to the foundation to give firmness and strength to the building [5–7]. In response, the use of a diagrid structural system has become widespread because of its aesthetic potential combined with its structural efficiency [6]. Nowadays, the growth of the diagrid structural system in both the industrial private sectors and the public sectors during the past decade is of interest to numerous researchers looking for the guidelines for the optimum structural system for multi-story buildings in terms of saving time and the cost of the construction [7]. The framing systems that are used in a building work by transferring all the building loads to the foundation to provide firmness and strength to the building [8]. The framing system is also critical to the building's loading path, leading to an appropriately designed and erected framing system [9]. Framing systems can be found in load-bearing walls, beams, girders and other elements of the structure. Steel moment, braced, and diagrid frames offer the strength that aids in lateral load resistance. Furthermore, these structures are known for their strength, durability, and cost-effectiveness. These structures are typically constructed with steel frames and modular components, such as walls, floors, and roofs, which are designed to fit together like a giant puzzle [10]. Therefore, the modules are then connected together with bolts and welds to form a structural unit. This construction method allows for a faster assembly process and a more efficient use of materials compared to the traditional construction methods. Additionally, these structures are often designed with fire-resistant elements, which can help to reduce insurance and/or construction costs. In recent years, diagrid systems have been used in the construction of multi-story buildings, including the world's tallest building, the Burj Khalifa in Dubai, (United Arab Emirates). Other examples of diagrid buildings include the Shanghai Tower (Shanghai, China), the Trump International Hotel and Tower in Chicago (IL, USA), and the Empire State Building in New York City (NY, USA) [11,12]. Diagrid systems have also been used in the construction of sports stadiums, such as the Mercedes-Benz Stadium in Atlanta (GA, USA) [13].

### 1.2. Literature Review

Diagrid structures made of steel are effective in providing strength and support for massive structures. For instance, the popular diagrid buildings around the globe include the Jinling Tower of China, the Cyclone Tower in Asia, the Capital Gate Tower in Abu Dhabi, United Arab Emirates, the Hearst Tower in New York, NY, USA and the Swiss Re in London, UK [14]. Moment frame systems, also called rigid frame systems, consist of beams and columns used to reinforce concrete and steel structures. Hence, the since moment frames are more flexible than shear walls and braced frames as showing in the Figure 1 [15]. In the past, the diagrid system, reinforced steel, and concrete members were used in the construction of tall buildings [16]. On the other hand, braced structures are reinforced by steel to increase the strength of the building. The steel diagrid system is one of the best structural systems for constructing tall buildings that are irregularly shaped. Moreover, the benefits associated with diagrid systems in the construction of structures include the improvement of the aesthetic view of the building, the minimizing of the use of steel by up to 20% as compared to other systems, the use of a simple construction technology instead of highly skilled labor, and the elimination of the need to have a structure with corner columns; the overall structure requires less space if designed correctly, and the materials weigh less while providing maximum strength [17]. In addition, the complexities associated with steel members include the size of the structure or the shape of the structure, which may require the use of certified pre-fabricated steel connections, which can lead to an increase in construction costs. Currently, there are limited experienced crews available to install diagrid structures because the system requires skilled professionals who can handle this type of structure. Braced structures have several advantages in that they are economical, easy to erect, and occupy less space. Moreover, these structures can resist seismic forces as well as offer flexibility in terms of strength and stiffness [18]. Furthermore, moment frames need less work and less details, but at the same time, they have an increase in lateral displacements in cases of natural catastrophes such as earthquakes. Several

researchers mentioned that the most basic structures that are used to resist both lateral and vertical loads are moment-resisting frames [19–22]. Moreover, the combination of the moment frame with non-structural element such as masonry increases the performance of the frames; for example, the Hotel San Diego (New York, NY, USA) is a six-story reinforced concrete infilled-frame structure that resists loads [23]. In 2007, Karavasilis found that braced frames have the extra vertical steel trusses that are very efficient in resisting lateral loads [24]. On the other hand, the bracing is a very effective global upgrading strategy as it enhances the global stiffness and strength of steel and composite frames [25]. The positioning of braces, however, can be problematic as they can interfere with the design of the façade and the position of openings. Braces can be aesthetically unpleasant where they change the original architectural features of the building [26,27].

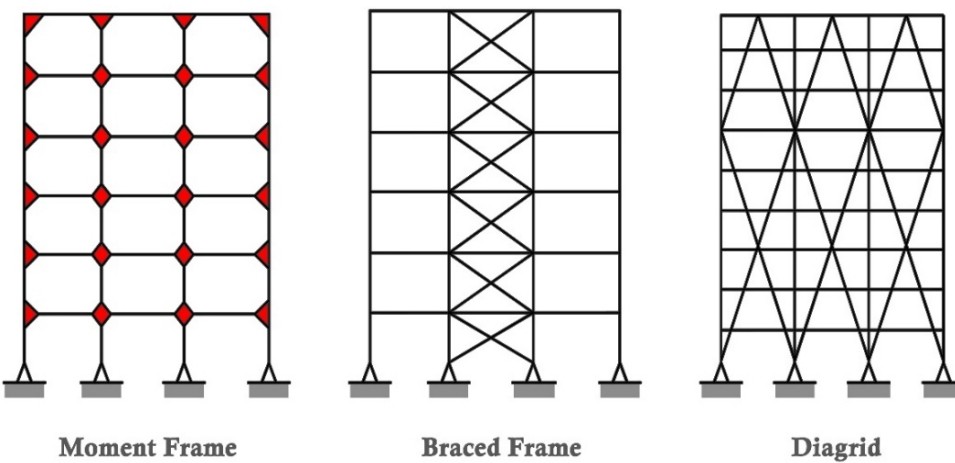

**Figure 1.** Moment frame vs. braced frame vs. diagrid structure.

Varkey and George investigated the reasons why diagrid structures have become very popular among architects and engineers; they assessed the efficiency and reliability of diagrid structures under gravity, wind, and seismic loads. Specifically, they provided their analysis on the performance of diagrid structures with regard to seismic and wind loads, gravity, and other parameters. Furthermore, the findings of the analysis in this study, such as story displacement and story drift, provide reliable factors behind the rapid growth of diagrid structures [28]. In 2013, Boake mentioned that the diagrids are one of the most innovative and adaptable milestones achieved in structural buildings [29]. Chatzikonstantinou studied the façade design as one of the modern constructions that require detailed and deliberate assessment of the performance of various designs; in addition, this study evaluates the performance of the various designs using differential evolution (DE) to compare the performance of two DE variants, using the hypervolume metric [30]. According to Moon in 2013, the number of floors loaded in a single module determines the height of the diagrid, which normally ranges from two to six floors. Hence, the optimal bending angle for rigidity is at about 90 degrees, and it is 35 degrees for the diagonals for shear rigidity as shown in the Figure 2 [31]. In addition, another study reveals that the normal acceptable range is normally 60–70 degrees [32].

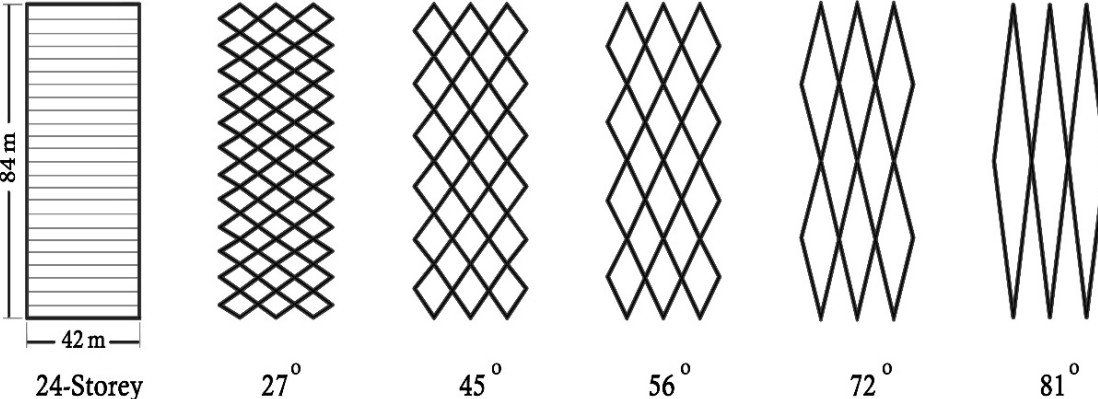

**Figure 2.** Diagrid angles in 24-story models.

### 1.3. Motivation of Research

Based on the rapid development in buildings in terms of structural analysis and analytical methods, as mentioned in the previous literature review, a few studies implemented different lateral framing systems in multi-storied buildings to find a better lateral framing system which reflects the critical problems in designing buildings, particularly the height, as inadequate design can expose buildings to excessive movements during earthquakes and wind. The motivation of the study is to investigate and analyze three lateral framing systems to find out which system is stiffer, lighter, and has the least steel requirement.

### 1.4. Objective of Research

The objective of this study is to investigate building designs; it covers four-story and eight-story buildings that employ different structural designs that are $96' \times 96'$ in the plane. Thus, the analyses are based on ASCE 7–10 loadings for the building designs. The RAM structural analysis software was used to model and analyze the structural systems. All the designs are for an office building located in Omaha, NE, USA.

## 2. Procedure of Work

The structural dimensions were considered to be four-story and eight-story buildings with a floor-to-floor height of 15 ft., and three bays of 32 ft. $\times$ 32 ft. in each direction. The space between the columns was 32 ft. The maximum building height was 120 ft. for the eight-story building and 60 ft. for the four-story building. The following Figures 3–5 show a typical floor plan and elevation for the models. The elements shown in red are the main lateral resisting members analyzed during this study. The RAM structural system is advanced software used for analyzing both multi-story buildings and high-rise structures. Hence, it was developed by Bentley Systems, Inc. (Exton, PA, USA) and is used by many engineers and architects. The software is designed to be used in the early stages of a project to analyze and design the structure and then to monitor the progress of the project throughout the construction process. The software has powerful features that allow users to quickly evaluate and analyze existing building structures, as well as to design new structures [33]. The models have been used for the analysis of shear, drift, and displacement results. Hence, the seismic and wind analysis parameters are also examined as the buildings located in the Omaha, NE, USA are considered for the proposed data. The load combination taken for the wind and seismic parameters is derived from the ASCE 7–10 [34].

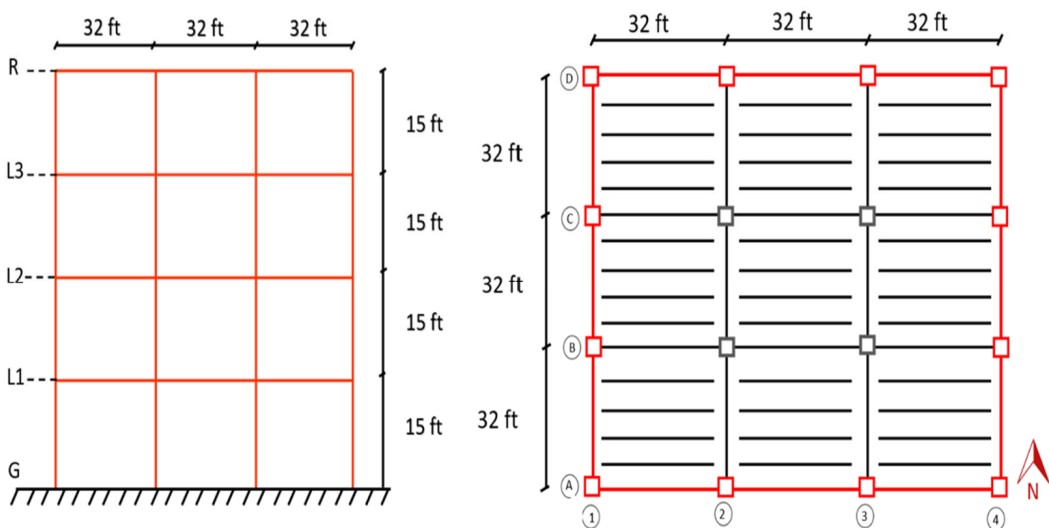

**Figure 3.** Typical floor plan and elevation of moment frame.

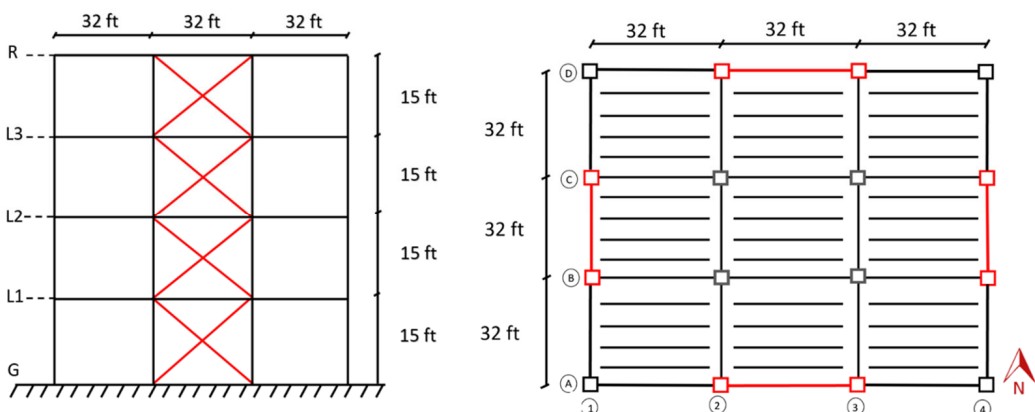

**Figure 4.** Typical floor plan and elevation of braced frame.

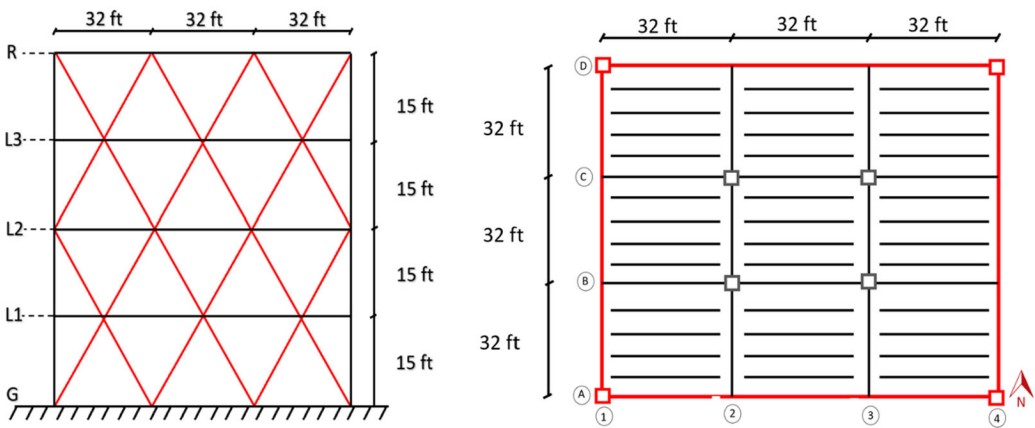

**Figure 5.** Typical floor plan and elevation of diagrid system.

*2.1. Data of Structural Loads*

2.1.1. Gravity Loads

Table 1 shows the details of the aforementioned structure input data to analyze the diagrid, braced, and conventional building frame.

**Table 1.** Gravity and snow loads input data.

| Gravity Dead Loads | | Snow Loads | |
|---|---|---|---|
| Component | Load (psf) | Description | Input |
| Mechanical/Electrical/Plumbing | 10 | Building Classification | II |
| Structure self-weight | As calculated | Exposure Factor, Ce | 0.9 |
| Ponding | 5 | Thermal Factor, Ct | 1 |
| Allowance for roofing system | 7 | Importance Factor, I_s | 1 |
| Exterior walls (window curtain wall) | 8 | Ground Snow Load, Pg | 25 psf |
| Partitions | 10 | Flat Roof Snow Load, pf | 15.75 psf |

2.1.2. Lateral Loads

The lateral loads were calculated and obtained using ASCE 7–10. These loads were divided into two sections: wind loads and seismic loads, as shown in Tables 2 and 3. The drift ratio under the design wind load, as defined by ASCE 7-88, is set at a 0.0025 drift (H/400) and is appropriate for a speculative office building [35]. Furthermore, the ASCE 7 and the other building codes did not mention any specific drift limits under wind loads, but again, the most commonly used wind drift limit is (H/400) for low-rise buildings.

The building structure includes complete lateral and vertical force-resisting systems capable of providing adequate strength, stiffness, and energy dissipation capacity to withstand the design ground motions within the prescribed limits of deformation and the strength demand.

**Table 2.** Wind loads input data.

| Wind Loads | |
|---|---|
| Wind Code | ASCE 7–10 |
| Risk Category | II |
| V, Basic Wind Speed | 120 mph |
| $K_d$, Wind Directionality Factor | 0.85 |
| Exposure Category | C |
| Building Height, h | 60 ft. |
| $K_z$, Velocity Pressure Coefficient | (ASCE 7–10) |
| $K_{zt}$, Topographic Factor | 1 |
| G, Gust Effect Factor | 0.85 |
| Enclosure Classification | Enclosed |
| $GC_{pi}$, Internal Pressure Coefficient | 0.18 |
| $C_p$, External Pressure Coefficient | (ASCE 7–10) |
| Windward | 0.8 |
| Leeward | −0.5 |
| Side Wall | −0.7 |

**Table 3.** Seismic loads input data.

| Seismic Loads | |
|---|---|
| **Design Parameters** | **Value** |
| $S_s$, mapped MCE response acceleration at short periods | 0.095 g |
| $S_1$, mapped MCE response acceleration at 1 s period | 0.045 g |
| $F_a$, short-period site coefficient | 1.6 |
| $F_v$, long-period site coefficient | 2.4 |
| $S_{MS}$, response acceleration at short periods | 0.152 g |
| $S_{M1}$, response acceleration at 1 s period | 0.108 g |
| $S_{DS}$, design response acceleration at short periods | 0.101 g |
| $S_{D1}$, design response acceleration at 1 s period | 0.072 g |

### 2.2. Design Verifications

These design verifications used the RAM structural system design for all the structural elements (beams, columns, and braces), and in order to validate the values found in RAM, they were checked using the following the design verifications.

### 2.2.1. Composite Steel Beam Verification

The composite steel beam and girder were checked based on AISC 360-10 for strength and serviceability [36]. The construction strength check compares the flexural and the shear capacity of the composite member. On the other hand, the construction serviceability checks the deflection of the composite members due to the self-weight of the steel; the camber design is also considered in this check. The following equation is used to obtain the total serviceability of the composite members:

$$y_{tot.} = (y_{max.}\,(_{cons.}) - Camber) + y_{max.}\,(_{DL}) + y_{max.}\,(_{LL}) \tag{1}$$

where $y_{max.}\,(_{cons.})$ is the maximum dead load deflection due to construction; $y_{max.}\,(_{LL})$ is the maximum live load deflection; $y_{max.}\,(_{DL})$ is the maximum dead load deflection; and the camber is $0.8 \times y_{max.}\,(_{cons.})$

### 2.2.2. Steel Column Verification

Three steel columns were checked in different locations in the building: the corner column, edge corner column, and center column. The values of $\varphi P_n$ found from RAM matched the values from the AISC manual checks. The following interaction equation was used, and the calculated value was less than 1, which means the selected columns were adequate.

$$P_r/P_c + 8M_r/9M_c < 1.0 \tag{2}$$

where $P_r$ is the required axial strength using the LRFD load combinations (kips); $P_c$ is the available axial strength (kips); $M_r$ is the required flexural strength (kip-inch); and $M_c$ is the available flexural strength (kip-in).

### 2.3. Response Modification Coefficient (R-Value)

The R-value for steel diagrid systems depends on the type of steel and the configuration of the system. Generally, steel diagrids with a combination of open and closed cells can have an R-value based on ASCE 7–10, where the response modification factor "R" is given for each seismic force resisting system. The R-value can be as high as 8 for ductile seismic framing systems. For this study, an R-value of six is used for the braced frame and eight for the steel moment frame. As previously stated, the R-value of the diagonal steel system is something that none of the current building codes addresses. However, in the study made

by Owings & Merrill LLP (Chicago, IL, USA) they estimated the seismic response factor R to be 3.64 [37].

## 3. Results and Analysis

The steel framing system used in a building works in union with the building's foundation to give firmness and strength to the building. The framing systems that resist lateral structural loads are the load-bearing walls, beams, girders, and diagonal framing braces of the structure. The moment, braced, and diagrid frames offer the strength that aids in lateral load resistance. This study compares the analysis results between the three lateral framing systems mentioned above in terms of wind, seismic load calculations, steel weight, displacement, story drift, and story shear.

The results collected from RAM are presented in charts and tables for comparison in terms of the three models. Finally, the design verifications for the composite steel structural are presented at the end.

### 3.1. Seismic Loads

The seismic load calculations were derived from three variables in this study, namely story force, height, and weight. The results of the three lateral framing systems are exhibited in Tables 4–6.

**Table 4.** Seismic load calculation (N-S) and (E-W) of moment frame.

| Story Level | Story Height (ft) | Height $h_x$ (ft) | Weight $W_x$ (kip) | $C_{vx}$ | Story Force $F_x$ (kip) | Story Shear (Calculated) (kip) | Story Shear (RAM) (kip) |
|---|---|---|---|---|---|---|---|
| Roof | 15 | 60 | 1083 | 0.41 | 23 | 0 | 0 |
| Level 3 | 15 | 45 | 1187 | 0.32 | 28 | 23 | 22 |
| Level 2 | 15 | 30 | 1187 | 0.19 | 11 | 51 | 52 |
| Level 1 | 15 | 15 | 1187 | 0.08 | 4 | 62 | 62 |
| Ground | 0 | 0 | 4644 | 0 | 0 | 66 | 66 |

Seismic base shear = 66 kips (RAM), 66 kips (calculated).

**Table 5.** Seismic load calculation (N-S) and (E-W) of diagrid system.

| Story Level | Story Height (ft) | Height $h_x$ (ft) | Weight $W_x$ (kip) | $C_{vx}$ | Story Force $F_x$ (kip) | Story Shear (Calculated) (kip) | Story Shear (RAM) (kip) |
|---|---|---|---|---|---|---|---|
| Roof | 15 | 60 | 1016 | 0.38 | 47 | 0 | 0 |
| Level 3 | 15 | 45 | 1120 | 0.31 | 38 | 47 | 46 |
| Level 2 | 15 | 30 | 1120 | 0.21 | 26 | 85 | 84 |
| Level 1 | 15 | 15 | 1120 | 0.10 | 12 | 111 | 109 |
| Ground | 0 | 0 | 4376 | 0 | 0 | 123 | 122 |

Seismic base shear = 122 kips (RAM), 123 kips (calculated).

**Table 6.** Seismic load calculation (N-S) and (E-W) of braced frame.

| Story Level | Story Height (ft) | Height $h_x$ (ft) | Weight $W_x$ (kip) | $C_{vx}$ | Story Force $F_x$ (kip) | Story Shear (Calculated) (kip) | Story Shear (RAM) (kip) |
|---|---|---|---|---|---|---|---|
| Roof | 15 | 60 | 1055 | 0.38 | 29 | 0 | 0 |
| Level 3 | 15 | 45 | 1158 | 0.31 | 24 | 29 | 29 |
| Level 2 | 15 | 30 | 1158 | 0.21 | 16 | 53 | 53 |
| Level 1 | 15 | 15 | 1158 | 0.10 | 8 | 69 | 69 |
| Ground | 0 | 0 | 4529 | 0 | 0 | 77 | 77 |

Seismic base shear = 177 kips (RAM), 177 kips (calculated).

The calculation typically considers various factors such as the magnitude and direction of the seismic forces, the height and mass of the building, and the design codes and standards that apply. The exact method of calculation may vary depending on the specific code or standard used, but the commonly used methods include static and dynamic analysis. The results in Figure 6 showed that the moment frame is the best system for high seismic zones as compared to the others because it has the requisite weight that can help sustain the building when under duress. The moment frame has the lowest base shear because it relies on the bending capacity of the beams and columns to resist lateral loads, while the braced and diagrid systems use a combination of compression and tension elements, such as braces or diagonals, to transfer the loads to the foundations. The overarching seismic base shear was set at 66 kip (calculated), with the RAM being 66 kips. The moment frame is expected to have the least seismic base shear based on the equations used because the system has an R-value of eight.

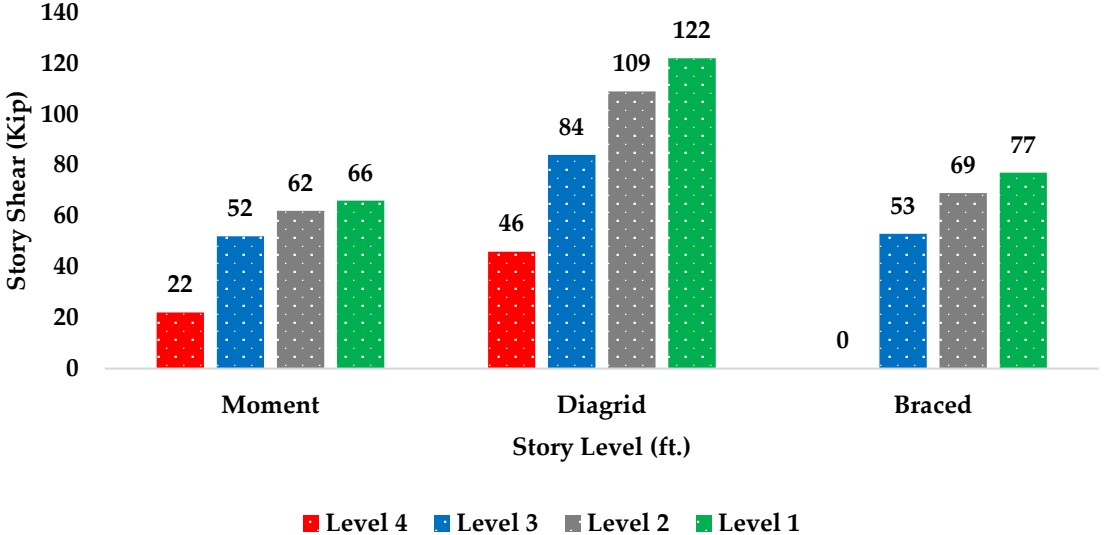

**Figure 6.** Seismic shear vs. story level.

*3.2. Inter-story Drift*

Inter-story drift is defined as the inherent displacement of a structure in a lateral direction, in which the relative level is examined against the level above and below it. The ratio for story drift is normally summated from the division of the story drift and the height of the story. Structural engineers and scholars have all been congruent on the fact that the allowable story drift as specified in ASCE 7–10 is $\delta \leq h/400$, where (h) is the story height. The results of the three lateral framing systems are displayed in Table 7.

**Table 7.** Inter-story drift of moment, diagrid, and braced frame.

| Story Level | Inter-Story Drift (in) | | | Limit $\delta \leq h/400$ (in) | Load Case | OK? |
|---|---|---|---|---|---|---|
| | Moment | Diagrid | Braced | | | |
| Level 4 | 0.31 | 0.0058 | 0.063 | 0.45 | Wind | ok |
| Level 3 | 0.42 | 0.014 | 0.074 | 0.45 | Wind | ok |
| Level 2 | 0.44 | 0.017 | 0.076 | 0.45 | Wind | ok |
| Level 1 | 0.37 | 0.016 | 0.067 | 0.45 | Wind | ok |

As shown in Figure 7 within the moment frame, an increase of 0.07 inches was noted from the first floor to the second floor in the *x*-direction, an influx of 0.02 inches from the second to the third floor, and then finally a decrease of 0.11 inches from the third to the fourth flow. The noted decrease in story drift was caused by the inherent resistance to lateral forces, with the rigid frame causing a bending moment in the joints and members.

The inter-story increased based on the member sizes because the heaviest members are chosen. The diagrid system has a lower inter-story drift of 0.0058 inches compared to the moment frame, which has a higher inter-story drift of 0.31 inches. This suggests that the diagrid system may be more effective in resisting lateral loads and maintaining stability during earthquakes or wind events. This is also because the moment frame members are designed based on serviceability and not strength, as with the braced frame and diagrid.

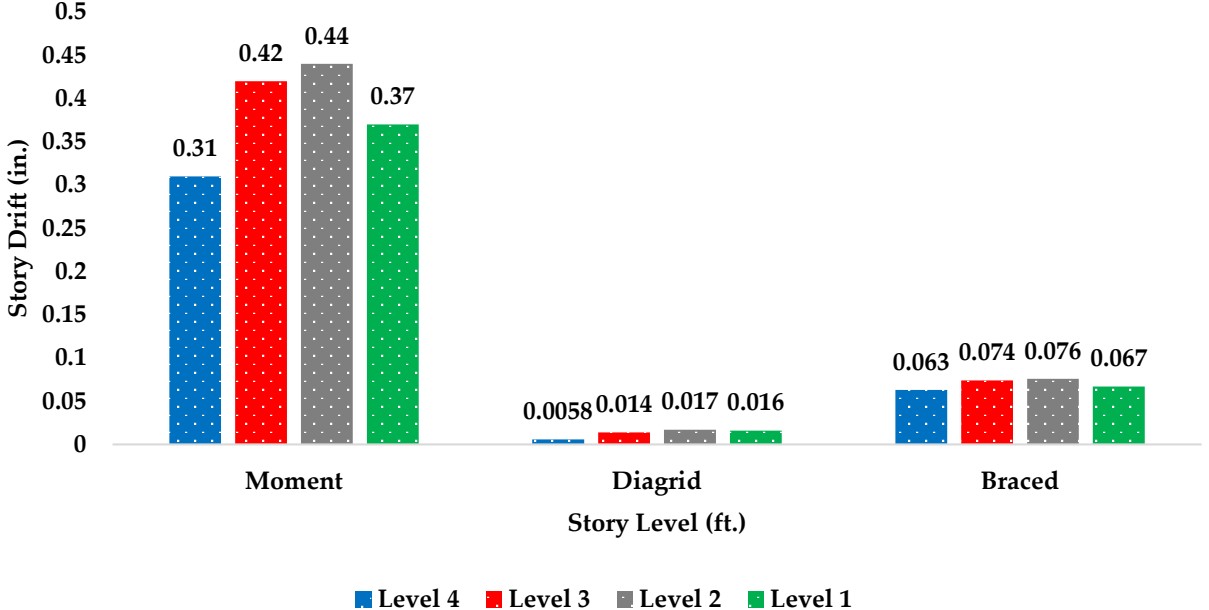

**Figure 7.** Inter-story drift vs. story level.

### 3.3. Building Displacement

The advent of technology over the years has helped in the development of architectural structures that are not only contemporaneous but also uphold the requisite strength standards. The primary analysis of the building displacement in the building indicated the limit of $\Delta \leq H/400$, where (H) is the height from the ground. The displacement in the x-direction increased on every floor for each framing system, as presented in Table 8, where it was trying to get close to the limit of 1.8 in.

**Table 8.** Building displacement of moment, diagrid, and braced frame.

| Story Level | Building Displacement (in) | | | Limit $\Delta \leq H/400$ (in) | Load Case | OK? |
|---|---|---|---|---|---|---|
| | Moment | Diagrid | Braced | | | |
| Level 4 | 1.53 | 0.045 | 0.28 | 1.8 | Wind | ok |
| Level 3 | 1.22 | 0.040 | 0.22 | 1.35 | Wind | ok |
| Level 2 | 0.81 | 0.031 | 0.14 | 0.90 | Wind | ok |
| Level 1 | 0.37 | 0.016 | 0.067 | 0.45 | Wind | ok |

The diagrid system is known for its high resistance to inelastic deformation due to its low displacement of 0.045 inches, which is achieved by a structural design focused on strength. On the other hand, the moment frame system as shown in Figure 8 has the highest displacement of 1.53 inches because it is designed based on serviceability, which involves ensuring the building's structural integrity under normal loads and accommodating for expected levels of deformation.

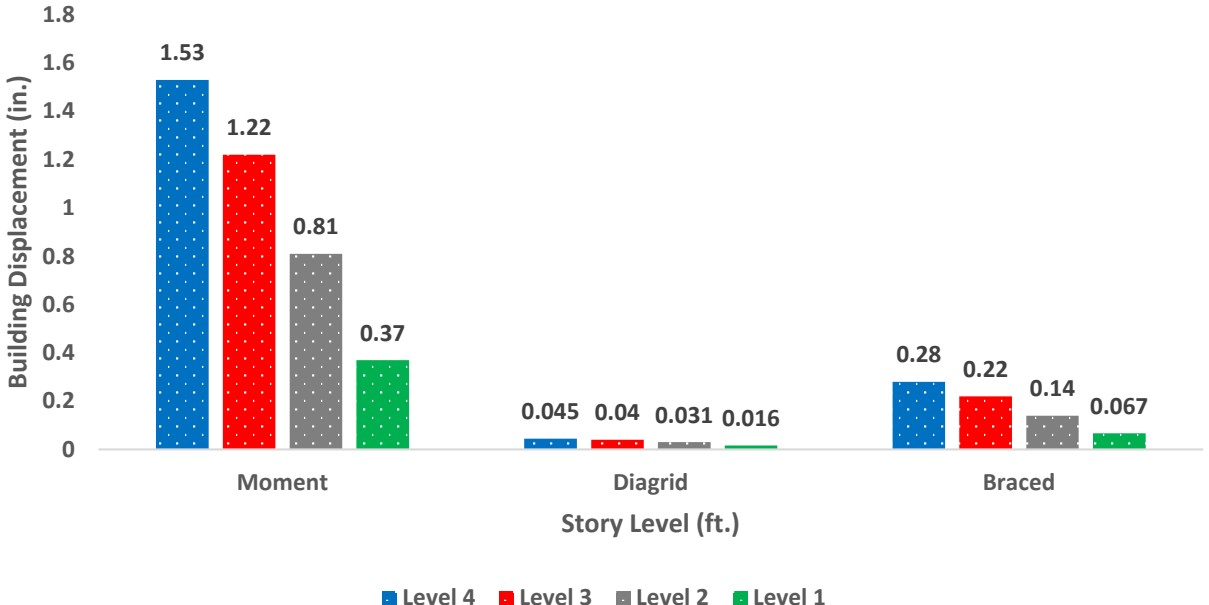

**Figure 8.** Building displacement vs. story level.

### 3.4. Story Stiffness

The story stiffness is defined as the inverse of the inter-story drift when a unit load is applied at that particular story only and is calculated as K = (story force/story displacement). All stories are rigid diaphragms. The displacement values for each story in the X, Y, and $\theta_z$ directions from RAM are taken at the center of mass. This check shows how much force is needed to push each story one inch. The following in Table 9 shows the story stiffness (K) for each of the framing systems.

**Table 9.** Story stiffness of moment, diagrid, and braced frames.

| Story Level | Story Stiffness | | |
|---|---|---|---|
| | *X*-Direction (kip/in) | | |
| | Moment | Diagrid | Braced |
| **Story 4** | 170 | 918 | 174 |
| **Story 3** | 210 | 1150 | 236 |
| **Story 2** | 299 | 1493 | 378 |
| **Story 1** | 676 | 2857 | 885 |

The inclined columns in the diagrid system helped in resisting any axial action imposed on the building. The analysis of story stiffness for the diagrid system represented in Figure 9 shows how stiff the system was as compared to the braced and moment frames, where the fourth story needed a force of 918 to push the story one inch, which was four times more force than the moment frame needed for the top story. In conclusion, the moment frame is the weakest frame compared to the other two, followed by the braced frame, and then the diagrid system.

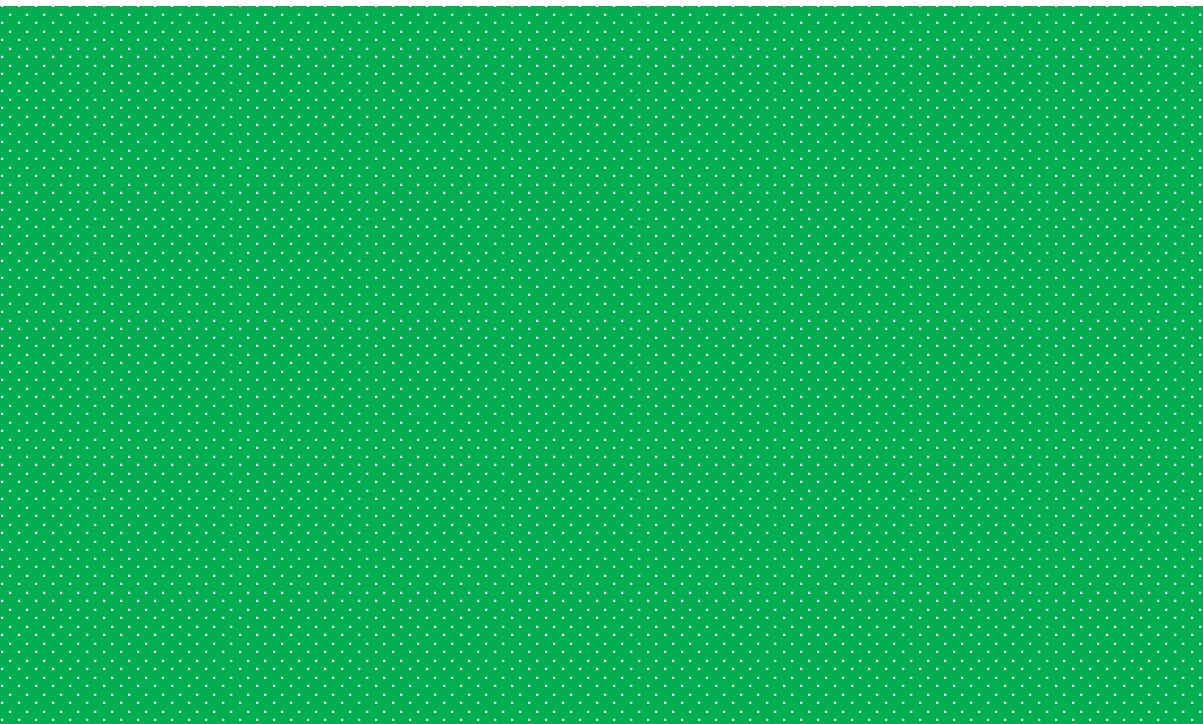

**Figure 9.** Story stiffness vs. story level (source: the author).

*3.5. Steel Weights*

   The summative conclusions are indicative of the fact that the moment, braced, and diagrid systems all come with different weight levels, as presented in Figure 10 below, with the moment frames having the highest total steel weight of 204 tons as compared to the 150 tons and 165 tons recorded for the braced and diagrid systems, respectively.

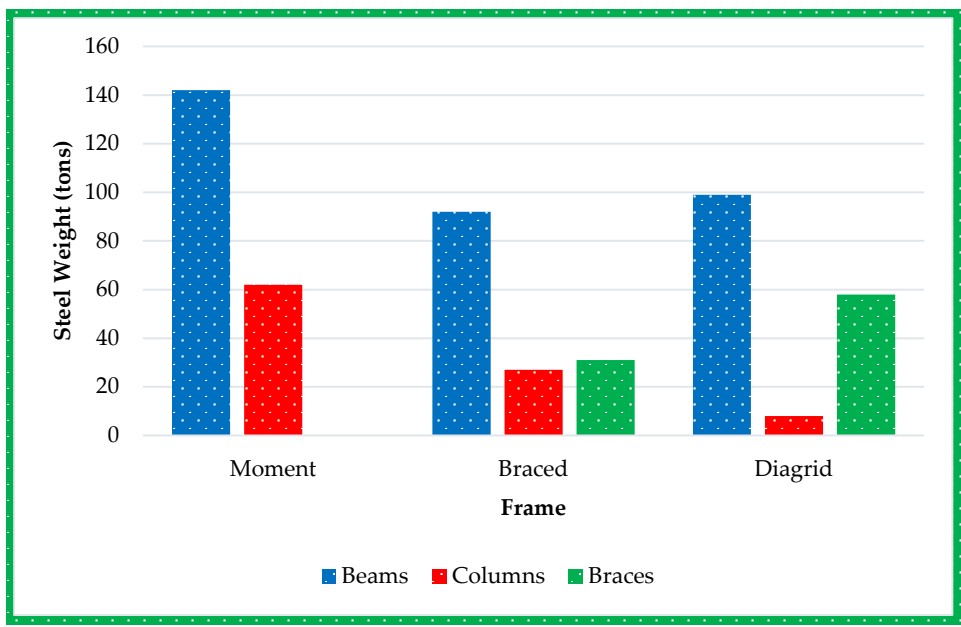

**Figure 10.** Steel weights of 4-story model (moment, braced, diagrid).

   Comparing the steel weights was the primary objective in this project in order to find out which system allows the savings of 20% of the structural steel, and according to the previous research, this can be conducted through the diagrid system. It is noted that the consumption of steel needed for the moment frame was more than that of the diagrid

system by 23%. On the other hand, the consumption of steel was 9% less in the braced frame as compared to that of the diagrid system. In conclusion, the use of the diagrid system instead of the moment frame allows a saving of 23% of the required steel, and no saving was found when using the diagrid instead of the braced frame.

### 3.6. Member Sizes

The moment frame system, steel beams, and columns are designed to resist bending and axial forces, respectively. The size of the beam and column will depend on the magnitude of the bending moment and the axial force in the structure. In the braced frame system, the steel members called braces are used to resist lateral forces such as wind and earthquakes. The diagrid system is a combination of both beam and bracing action as the diagonal members provide both strength and stability. The size of the diagonals will depend on the magnitude of the forces in the structure and the material properties of the steel. All the moment frame members were designed by serviceability rather than strength; so, the lightest members were chosen and the selection was limited to W18–W24 to get close to the total building displacement of 1.8 in., ignoring the connection because the goal of the study was to obtain lower steel weight, as shown in Tables 10–12. On the other hand, the braced frame and diagrid were designed for strength rather than serviceability because these systems are stronger than the moment frame in resisting the lateral loads of tension and compression.

**Table 10.** Braced frame member sizes (lateral and gravity).

| Braced Frame Story Level | Gravity Beams | # | Columns | # | Lateral Beams | # | Columns | # | Braces | # |
|---|---|---|---|---|---|---|---|---|---|---|
| Story 4 | W8X10 W16X26 W21X44 W24X62 | 33 4 4 6 | W10X33 | 8 | W12X106 | 4 | W10X33 W10X49 | 4 4 | W10X54 | 8 |
| Story 3 | W8X10 W16X26 W21X44 W24X62 | 33 4 4 6 | W10X39 W10X49 | 4 4 | W12X106 | 4 | W10X39 W10X49 W10X54 | 4 2 2 | W10X54 | 8 |
| Story 2 | W8X10 W16X26 W21X44 W24X62 | 33 4 4 6 | W12X50 W12X58 | 4 4 | W12X106 | 4 | W10X49 W10X53 W12X53 | 3 4 1 | W10X54 | 8 |
| Story 1 | W8X10 W16X28 W21X44 W24X62 | 33 4 4 6 | W12X72 W12X67 | 4 4 | W12X106 | 6 6 | W10X54 W10X60 W12X68 | 2 2 4 | W10X54 | 8 |

Total steel weight = 150 tons.

**Table 11.** Moment frame member sizes (lateral and gravity).

| Moment Frame Story Level | Gravity Beams | # | Columns | # | Lateral Beams | # | Columns | # |
|---|---|---|---|---|---|---|---|---|
| Story 4 | W8X10 W24X62 | 33 6 | W12X50 | 4 | W18X35 W21X93 | 6 6 | W18X46 W18X86 | 6 6 |
| Story 3 | W8X10 W24X62 | 33 6 | W12X58 | 4 | W24X94 W24X104 | 6 6 | W18X65 W24X117 | 4 8 |
| Story 2 | W8X10 W24X62 | 33 6 | W12X72 | 4 | W24X117 W24X131 | 6 6 | W21X111 W24X131 | 4 8 |
| Story 1 | W8X10 W24X62 | 33 6 | W12X87 | 4 | W24X117 W24X146 | 6 6 | W24X146 W24X162 | 4 6 |

Total steel weight = 204 tons.

**Table 12.** Diagrid system member sizes (lateral and gravity).

| Diagrid System | | | | | | | | | | |
|---|---|---|---|---|---|---|---|---|---|---|
| Story Level | Gravity Beams | # | Columns | # | Lateral Beams | # | Columns | # | Braces | # |
| Story 4 | W8X10 | 33 | W12X50 | 4 | W10X33 | 2 | — | 0 | W12X40 | 12 |
| | W24X62 | 6 | | | W12X72 | 2 | | | W12X53 | 12 |
| Story 3 | W8X10 | 33 | W12X58 | 4 | W12X72 | 2 | — | 0 | W12X40 | 12 |
| | W24X62 | 6 | | | W14X99 | 2 | | | W12X53 | 12 |
| Story 2 | W8X10 | 33 | W12X72 | 4 | W12X40 | 2 | — | 0 | W12X53 | 12 |
| | W24X62 | 6 | | | W12X72 | 2 | | | W12X65 | 12 |
| Story 1 | W8X10 | 33 | W12X87 | 4 | W12X72 | 2 | — | 0 | W12X65 | 12 |
| | W24X62 | 6 | | | W12X89 | 1 | | | W12X72 | 12 |
| | | | | | W24X68 | 1 | | | | |

Total steel weight = 166 tons.

### 3.7. Four-Story vs. Eight-Story Models

This section shows the modeling when doubling the height of the three models (moment, braced, and diagrid) to see how the height of the building affects the lateral systems. The height went from 60 ft. to 120 ft. above the ground, as shown in the Figures 11–14, with same building dimensions and the considered loads. The comparisons are discussed in the conclusion in terms of seismic base shear, displacement of the top story, steel weight, and story stiffness.

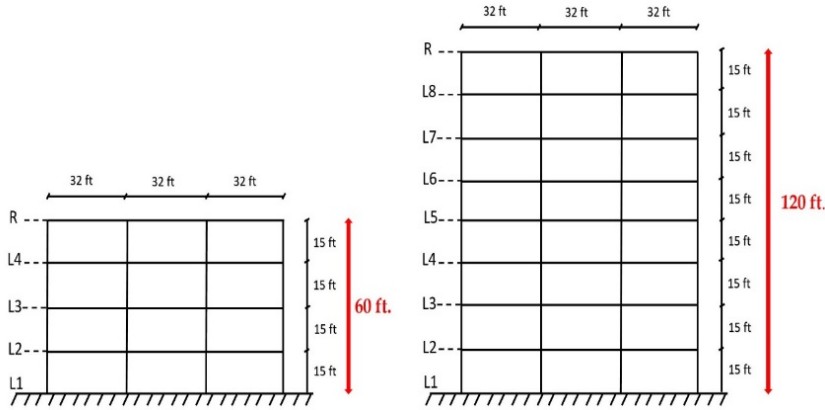

**Figure 11.** Elevations of 4-story and 8-story models.

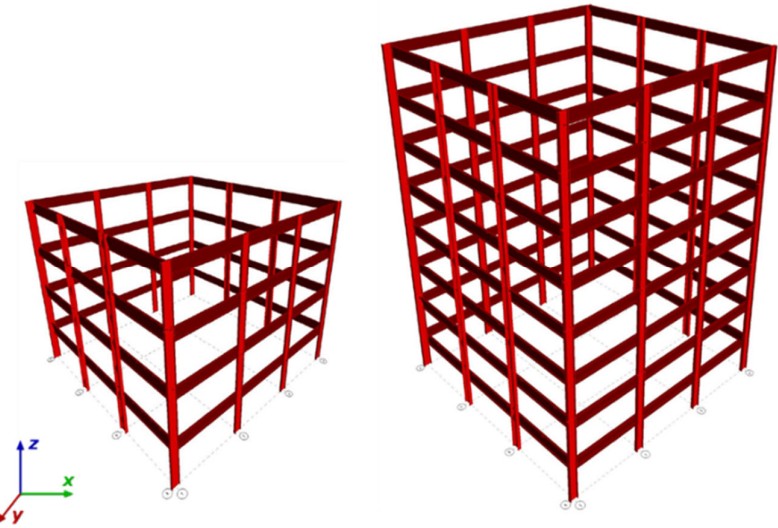

**Figure 12.** Moment frame of 4-story vs. 8-story model.

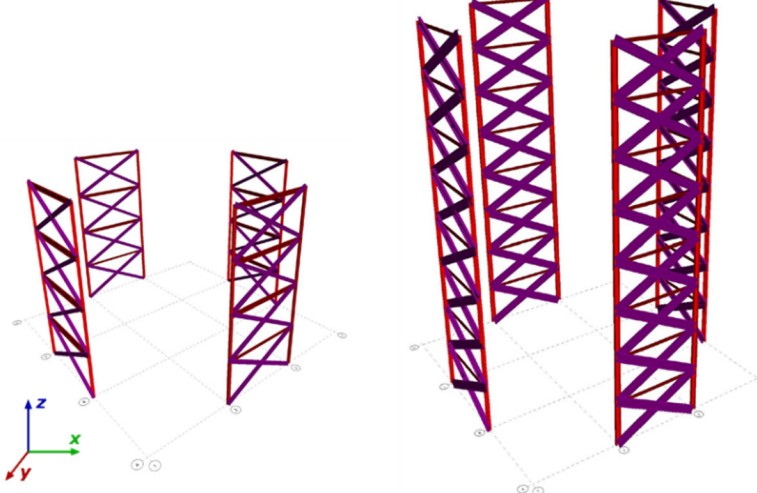

**Figure 13.** Braced frame of 4-story vs. 8-story model.

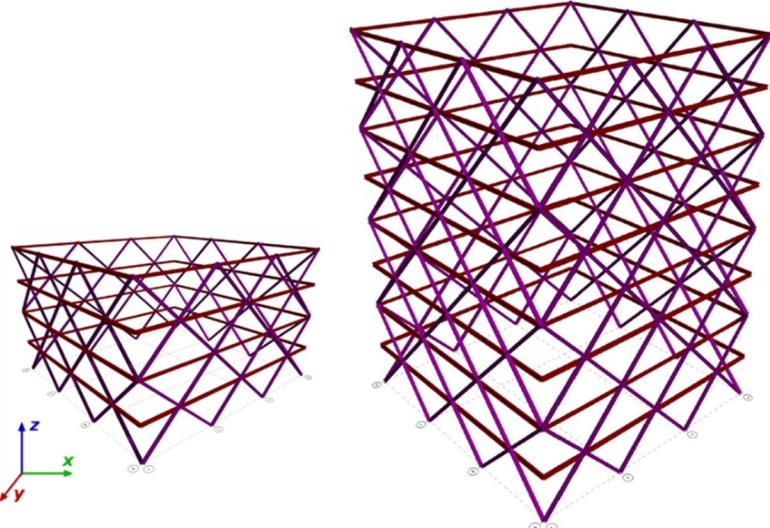

**Figure 14.** Diagrid system of 4-story vs. 8-story models.

## 4. Discussion

In this paper, the analysis and design of the three lateral framing systems were presented in detail as four-story and eight-story buildings that were $96' \times 96'$ in the plane and utilized moment frames, braced frame, and diagrid framing structural systems. The RAM structural system was used to analyze the frames, considering all the load combinations. The wind and seismic loads with all the structural elements were designed using ASCE 7–10. This section presents the discussion of the analysis and the results between the three lateral framing systems in terms of wind and seismic loads, building lateral displacements, the lowest of which was 0.045 inches, story drift, and story shear for each structural system, and then studies the steel framing weights for each system.

The diagrid showed better values than the braced and moment frame as a lateral load resistance system; in addition, as the building systems were kept symmetrical, the seismic base shear was the same in both the lateral and the transverse directions. The seismic shear values were based on the R-value found in the ASCE 7–10 along with the equations as a higher R-value will result in a lower base shear. The diagrid system had the highest seismic base shear of 122 kips because it had an R-value of 3.64 when compared to the braced frame and moment frame; this made the system stiffer in comparison to the other two.

From the values found in RAM for the displacement and story drift, it is noted that, as the story height increased, the displacement and story drift of the building increased. All

the values were within the code maximum drift/displacement limits for all three systems, where higher values were noticed in the moment frame when compared to the other two systems; this makes the moment frame a flexible system and suitable for high seismic zones.

The wind base shear control was 176 kips for all the systems even though the highest seismic base shear was 122 kips, as found in the diagrid system. All the models were assumed to be in Omaha, where the wind loads were expected to dominate because Omaha is not considered to be in a high seismic zone.

Investigating the steel weights was the primary objective of this project in order to find out which system allows a saving of 20% of the structural steel, as mentioned in the literature review. It is noted that the consumption of steel needed for the moment frame was more than that of the diagrid system by 23%. On the other hand, the consumption of steel was 9% less in the braced frame than it was in the diagrid system. Therefore, using the diagrid system instead of the moment frame allows a saving of 23% of the required steel. The following diagram that represented in Figure 15 shows the final comparison results between the three systems with the high and low values of each term.

| | Seismic Base Shear (Kip) | Top Building Displacement (in.) | Top Interstory Drift (in.) | Steel Weights (tons) | Top Story Stiffness (Kip/in.) |
|---|---|---|---|---|---|
| **Increase/Decrease to basis** ↑ | +58% **Diagrid** (122) | +400% **Moment** (1.53) | +392% **Moment** (0.31) | +36% **Moment** (204) | +427% **Diagrid** (918) |
| **(Basis value)** | 1.00 **Braced** (77) | 1.00 **Braced** (0.28) | 1.00 **Braced** (0.063) | 1.00 **Braced** (150) | 1.00 **Braced** (174) |
| **Increase/Decrease to basis** ↓ | -14% **Moment** (66) | -83% **Diagrid** (0.045) | -90% **Diagrid** (0.0058) | +10% **Diagrid** (165) | -2% **Moment** (170) |

**Figure 15.** The final comparison between the systems based on a 4-story model.

Another comparison was carried out for each system to investigate the changes after doubling the height of each building from 60ft. to 120ft., using the same loads, assumptions, and the same comparisons terms.

The seismic base shear doubled for all the systems, which was an expected outcome after increasing the height of the building. The top building displacement found in the 8-story building increased by 0.22 in. in the diagrid; 1.87 in. in the moment; and 2.22 in. in the braced frame, which means that the height of the building is important in the design because the higher building displaced more as in comparison to the short building, which makes it less stiff.

In terms of the story stiffness, the top story stiffness for all the systems decreased and became less stiff; for example, the top story found in the diagrid was 918 kip/in. and went down to 217 kip/in after the doubling of the height; from 170 kip/in to 76 kip/in for the moment; and from 174 kip/in to 78 kip/in in the braced frame.

The final comparison was in terms of the changes in steel weight; the moment and braced frames increased by more than two times when they were required to have bigger member sizes for the 8-story building. On the other hand, the diagrid system increased only by less than one time and kept the same member sizes in the 4-story building. Therefore, diagrid systems make the best approach, followed by the braced frame, and then the moment frame in terms of steel weight saving.

## 5. Conclusions

This study was conducted to achieve the optimal lateral framing system by studying the parameters, which included wind, seismic loads, building lateral displacements, story

drift, story stiffness, and story shear for each structural system. Six different building models were investigated, and it is concluded from the results in the discussions that the diagrid systems make the best approach, followed by the braced frame, and then the moment frame, with regard to steel weight. The diagrid system is worth more future research because it is a new innovative structural system that can help industries today. The conclusions are surmised in the following:

- Better lateral load resistance system when compared to braced and moment frames.
- The diagonal members on the periphery of the structure resist both the lateral and gravity loads, which makes the system more effective.
- Stiffer system than braced and moment frame.
- Stiffer system even with high-rise building when compared to moment and braced frame.
- Diagrid without corner columns is not very different from the ones with corner columns, but it can save 6% more of the required steel.
- The number of structural elements required (exterior vertical columns eliminated) in the building is reduced, which gives the building a more aesthetic look, especially in the interior space.

**Author Contributions:** Conceptualization, H.A.D.; validation, S.M.; formal analysis, H.A.D. and M.F.A.-K.; investigation, M.F.A.-K.; resources, J.A.L.; data curation, M.A. and M.F.A.-K.; writing—original draft, H.A.D.; writing—review and editing, M.A.; visualization, S.M. and N.I.M.Y.; supervision, J.A.L., N.I.M.Y. and A.M.; funding acquisition, A.M. All authors have read and agreed to the published version of the manuscript.

**Funding:** The University of Nizwa supports paying this publication's article processing charges (APC).

**Institutional Review Board Statement:** Not applicable.

**Informed Consent Statement:** Not applicable.

**Data Availability Statement:** All data used in this research can be provided upon request.

**Acknowledgments:** The authors would like to thank the University of Nizwa for its financial support.

**Conflicts of Interest:** The authors declare no conflict of interest.

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
