# Peer review of "Analysis and Design of Lateral Framing Systems for Multi-Story Steel Buildings"

_2673-3161, doi:10.3390/applmech4020022_

Round 1

Reviewer 1 Report

In this paper, the authors have conducted to determine the optimal lateral framing system for each structural system by studying a number of parameters, including wind, seismic loads, building lateral displacements, story drift, story stiffness, and story shear.

 The topic of the paper is interesting, but the authors need to address the following comments:

1.  The introduction to a research paper is where you set up your topic and approach for the reader. It has several key goals: Present your topic and get the reader interested. Provide background or summarize existing research. Position your own approach. The authors should revise this section and also cite the following papers:

https://doi.org/10.1016/j.jobe.2022.104831

https://doi.org/10.1016/j.istruc.2022.10.106

https://doi.org/10.3390/su142012990

https://doi.org/ 10.1061/JAEIED.AEENG-1491

3. The authors should add a section called “Verification Study” and discuss the related issues.

 4. Why the authors have not conducted nonlinear analysis?

5. How did the authors make sure the braces buckle during the analysis? Did they model any imperfection?

5.  What design code has been used?

6.  What practical help does this research give to engineers in design offices?

Author Response

Thank you for your positive, fruitful comments and suggestions, which have improved our manuscript's quality. Please find below is the revision report for your attention and perusal. The responses have been arranged based on your feedback in the review process.

Reviewer 2 Report

The authors reveal a deep understanding of the analyzed issue and have carried out intensive work, and the obtained results might be useful; hence, the paper could be considered for publication. However, the paper cannot be published in its present form, mainly because the explanations are extraordinarily obscure. The authors are suggested to resubmit their work after addressing completely the following observations.   

1. The logic of the abstract needs to be further improved.

2. The introduction is way too lengthy. Please shorten this part.

3.  The displacement presented in figure 8 is also unrealistic for the kind of heights of the buildings.

4.  The reasons for selecting four and eight-story models should be specified.

5.  There are different types of bracing systems. Why only X-bracing systems selected and provided only at central bays of the building model? It can be provided at 2 bays near the corners; in that case, analysis results may be different.

6.  What angle of diagrid is considered in the analysis or in the analysis diagrid is covering two stories at a time, it can cover only one or more stories.

7.   In section 3.1, for the explanation of Figure 6 it was interpreted by the author that, “An increase in the story height was noted to cause an increase in its weight and a decrease in its story force for the moment frame”. This is not appropriate to explain why the moment frame showed lower base shear than braced and diagrid models. Another thing is that increase in seismic weight directly results in an increase in base shear and story shear results.

8.  It is shown in the study that the steel weight of the moment frame comes as 204 tons which is more than the other two systems viz. diagrid and braced. As seismic base shear is directly proportional to seismic weight, accordingly base shear in the case of moment frame should be more but results in the study indicate that base shear for the diagrid model is more (122 Kip). These results are contradicting each other.

9.  As mentioned by the author in the discussion section, “Investigating the steel weights was the primary objective of this project to find out which system allows a saving of 20% of the structural steel as mentioned in the literature review”. If it is already discovered in past studies that a diagrid structure results in 20% savings in steel consumption, in that case, an explanation should be required about the main objective of the study.

Author Response

(The authors gave the same response as above.)

Reviewer 3 Report

This manuscript presents a numerical study on the comparison between three different structures for multi-story steel frame buildings. This work may be interesting for a specific field, but it is requires further improvement regarding the justifications of findings and other modifications before it can be further evaluated.

At first, the Abstract section should be revised regarding language errors. Moreover, they should also mention some findings of their research in this section.

In the Introduction section, the authors should add more references regarding similar studies on the analysis of the behavior of different structures under seismic, wind and snow loads. Moreover, the novelty should be more clearly stated and justified.

More information about the RAM software should be presented and the equations solved by the model should be directly mentioned. Regarding the seismic and snow loads, the authors should mention whether they are obtained from the relevant literature and justify their suitability. Especially, for the snow loads the authors did not mention them in the Results and Analysis section.

Figures 6-10 should be better changed to bar charts as floors are not a continuous variable and it would be also better to display floors on the x axis and the calculated quantity on the y axis. Furthermore, there is no Figure 9, thus Figure 10 should be numbered as 9 etc.

In Tables 4-6, the authors present calculated results by some method which is not mentioned and compare it to the results obtained by the RAM software. More details should be provided about these calculations.

In some parts of the manuscript an "R-value" is mentioned but no other details are presented and its values are not presented for every case.

In subsections 3.2, 3.3 and 3.6 the authors should justify the findings in more detail apart from just mentioning that some designs are used for strength and others for "serviceability". 

Author Response

(The authors gave the same response as above.)

Round 2

Reviewer 1 Report

The answers to the third and fourth questions of this referee are appropriate, but they should be added in the text of the article. According to the first comment, they authors should add this reference:

https://doi.org/10.3390/su142012990

Author Response

1-The answers to the third and fourth questions of this referee are appropriate, but they should be added in the text of the article. According to the first comment, they author should add this reference:

https://doi.org/10.3390/su142012990

Response: we would like to thank you for the valuable comments and the reference have been added as the reviewer suggested.

Reviewer 2 Report

The Revisions are satisfactory in my opinion, and I would certainly recommend the Editors to Publish the Paper in their esteemed Journal.

Thank You.

Author Response

Comments: The Revisions are satisfactory in my opinion, and I would certainly recommend the Editors to Publish the Paper in their esteemed Journal.

Response: Thank you very much for your valuable feedback and suggestions. We truly appreciate your help in making our manuscript better. and we thank you for taking the time to review our manuscript.

Reviewer 3 Report

The authors have conducted most of the necessary modifications to their manuscript. Thus, it can now be accepted for publication.

Author Response

Comments: The authors have conducted most of the necessary modifications to their manuscript. Thus, it can now be accepted for publication.

Response: Thank you very much for your valuable feedback and suggestions. We truly appreciate your help in making our manuscript better. and we thank you for taking the time to review our manuscript.
